# Induction of an early IFN-γ cellular response and high plasma levels of SDF-1α are inversely associated with COVID-19 severity and residence in rural areas in Kenyan patients

Perpetual Wanjiku[1]*, Benedict Orindi[1], John Kimotho[1], Shahin Sayed[2], Reena Shah[2], Mansoor Saleh[2], Jedidah Mwacharo[1], Christopher Maronga[3], Viviane Olouch[2], Ann Karanu[2], Jasmit Shah[4], Zaitun Nneka[2], Lynette Isabella Ochola-Oyier[1,5], Abdirahman I. Abdi[1,5], Susanna Dunachie[6], Philip Bejon[1,7], Eunice W. Nduati[1,5], Francis M. Ndungu[1,5]*

1 Centre for Geographic Medicine Research (Coast), Kenya Medical Research Institute (KEMRI)-Wellcome Trust Research Programme, Kilifi, Kenya, 2 Aga Khan University Hospital, Nairobi, Kenya, 3 Nuffield Department of Population Health, Cancer Epidemiology Unit, University of Oxford, Oxford, United Kingdom, 4 Brain and Mind Institute and Department of Medicine, Aga Khan University, Nairobi, Kenya, 5 Nuffield Department of Medicine, University of Oxford, Oxford, United Kingdom, 6 Nuffield Department of Medicine, Centre for Global Health Research, University of Oxford, Oxford, United Kingdom, 7 Nuffield Department of Medicine, Modernising Medical Microbiology, University of Oxford, Oxford, United Kingdom

* pwanjiku@kemri-wellcome.org (PW); fndungu@kemri-wellcome.org (FMN)

## Abstract

### Introduction

COVID-19 was less severe in Sub-Saharan Africa (SSA) compared with Europe and North America. It is unclear whether these differences could be explained immunologically. Here we determined levels of *ex vivo* SARS-CoV-2 peptide-specific IFN-γ producing cells, and plasma cytokines and chemokines over the first month of COVID-19 diagnosis among Kenyan COVID-19 patients from urban and rural areas.

### Methods

Between June 2020 and August 2022, we recruited and longitudinally monitored 188 COVID-19 patients from two regions in Kenya, Nairobi (urban, n = 152) and Kilifi (rural, n = 36), with varying disease severity – severe, mild/moderate, and asymptomatic. IFN-γ secreting cells were enumerated at 0-, 7-, 14- and 28-days post diagnosis by an *ex vivo* enzyme-linked immunospot (ELISpot) assay following *in vitro* stimulation of peripheral blood mononuclear cells (PBMCs) with overlapping peptides from several SARS-CoV-2 proteins. A multiplexed binding assay was used to measure levels of 22 plasma cytokines and chemokines.

**Data availability statement:** All data files are available from the Harvard Dataverse at the KWTRP Research Data Repository https://doi.org/10.7910/DVN/XMTCTW.

**Funding:** This study/project is funded by the EDCTP2 Programme (grant number RIA2020EF-3042) which is supported by the European Union, and the Swedish International Development Cooperation Agency. https://www.edctp.org/projects-2/edctp2-projects/mobilisation-funding-covid-19-research-sub-saharan-africa/ FMN received this funding SD is supported by an NIHR Global Health Research Professorship (NIHR300791). The funders had no role in study design, data collection and analysis, decision to publish, or preparation of the manuscript.

**Competing interests:** The authors have declared that no competing interests exist.

## Results

Higher frequencies of IFN-γ-secreting cells against SARS-CoV-2 spike peptides were observed on the day of diagnosis among asymptomatic compared to patients with severe COVID-19. Higher concentrations of 17 of the 22 cytokines and chemokines measured were positively associated with severe disease, particularly interleukin (IL)-8, IL-18 and IL-1ra ($p < 0.0001$), while a lower concentration of SDF-1α was associated with severe disease ($p < 0.0001$). Concentrations of 8 and 16 cytokines and chemokines including IL-18 were higher among Nairobi asymptomatic and mild patients compared to their respective Kilifi counterparts. Conversely, concentrations for SDF-1α were higher in rural Kilifi compared to Nairobi ($p = 0.012$).

## Conclusion

In Kenya, as seen elsewhere, pro-inflammatory cytokines and chemokines were associated with severe COVID-19, while an early IFN-γ cellular response to overlapping SARS-CoV-2 spike peptides was associated with reduced risk of disease. Living in urban Nairobi (compared with rural Kilifi) was associated with increased levels of pro-inflammatory cytokines and chemokines.

## Introduction

Despite widespread transmission of SARS-CoV 2, SSA experienced a reduced burden of severe coronavirus disease 2019 (COVID-19) and associated mortality than North America and Europe [1,2]. This observation is both puzzling [3] and paradoxical [4], because the opposite was predicted by public health experts, fearing that the weaker and underfunded health systems in SSA countries would collapse when presented with similar rates of severe disease as initially reported in Europe. Some of the proposed, but still unproven, explanations for the reduced burden of severe COVID-19 in SSA include under-diagnosis of cases and mortality, a younger population [5,6], warmer climatic conditions with outdoor living, high levels of pre-existing cross-protective antibodies and T-cells induced from a high prevalence of infectious agents with SARS-CoV-2 like immune determinants, and immune regulation associated with either prior BCG vaccination [6–12] or chronic infections including helminths [13] and malaria [14]. However, the widespread exposure to SARS-CoV-2 and high rates of asymptomatic infection observed in SSA countries suggest that non-biological explanations are unlikely to fully account for the reduced rates of severe disease and mortality in the continent, relative to wealthier regions like Europe and North America [2,15]. Nevertheless, the underlying immunological mechanisms remain understudied.

Like other viruses, SARS-CoV-2 induces a broad range of pro-inflammatory cytokines and chemokines [16] that play key roles in either conferring protective immunity, or exacerbating immunopathology [17]. Notably, the pathogenesis of COVID-19 has been linked to dysregulated and excessive cytokine and chemokine responses

upon SARS-CoV-2 infection [18]. Numerous studies have linked increased levels of cytokines and chemokines to severe COVID-19 and associated mortality, including IL-1β, IL-1ra, IL-2, IL-6, IL7, IL-8, IL-18, IFN-γ, TNF-α, IFN-γ-inducible protein 10 (IP-10), granulocyte macrophage-colony stimulating factor (GM-CSF), monocyte chemoattractant protein-1 (MCP-1), and Macrophage inflammatory protein-1 alpha (MIP-1-α) [19–26]. Collectively called a "cytokine storm", a dysregulated cytokine response is implicated as the cause of the multiple organ failures and the acute respiratory distress syndrome (ARDS), which characterises severe COVID-19 and associated fatalities [27]. Thus, potential population differences in the context of cytokine and chemokine regulation could explain the differential burden in severe disease. Alternatively, there could have been a higher prevalence of pre-existing cross-protective antibody and T cell responses in SSA during the COVID-19 pandemic, owing to prior priming of the immune systems by increased exposure to pathogens with shared antigenic determinants with SARS-CoV-2.

Published data show that the characteristic antibody response to SARS-CoV-2 infection reported in other populations [28], where levels increase with time, viral loads and COVID-19 severity, were also experienced in Kenyan patients [29]. Thus, there is no evidence that the primary acute antibody response to SARS-CoV-2 in unvaccinated patients is sufficient for immune protection, without the contribution of other molecular and cellular players. Aside from antibodies, T-cell responses can control viremia, either directly by killing virus infected cells or indirectly by providing the relevant co-stimulatory molecules for supporting antibody production by B cells [30]. However, there is a paucity of data on the cellular response to SARS-CoV-2 in Kenyan patients, and its possible role in modulating disease severity. Associations between severe COVID-19 and elevated levels of pro-inflammatory cytokines like IFN-γ, TNF-α, and IL-17 and anti-inflammatory cytokines including IL-6 and IL-10, have been reported [31]. Furthermore, Samandari et al. [32] demonstrated the presence of SARS-CoV-2-specific T cell responses among asymptomatic Kenyans, pointing to a possible role of cellular immunity in protection against severe disease.

Together, the above studies highlight the importance of characterizing cellular immune responses in the local context to better understand the role of differential cytokine, chemokine, and T cell responses with disease outcomes. Furthermore, it is unclear whether the differences in the rates of severe disease between urban and rural dwellers within SSA, as well as between developed countries and SSA, could be explained immunologically. In this study, we collected longitudinal blood samples from patients from Nairobi (urban) and Kilifi (rural) with varying degrees of COVID-19 severity (asymptomatic, mild/moderate and severe) and compared levels of their *ex vivo* SARS-CoV-2 Spike peptide-specific IFN-γ producing cells, and levels of plasma cytokines and chemokines over their first month of COVID-19 diagnosis. We then investigated relationships between the magnitudes of cellular IFN-γ, and several plasma cytokine and chemokine response profiles, with well characterized clinical disease phenotypes and geographic locations of the respective patients.

## Methods

### Study design, setting and participants

Participant sampling has been described previously [29]. Briefly, we included 400 blood samples from a longitudinal cohort study of 188 patients that aimed at understanding the kinetics of naturally acquired immune responses to SARS-CoV-2 among COVID-19 patients from two sites in Kenya: 1) The Aga Khan University Hospital (AKUH) in Nairobi, an urban metropolitan academic medical Centre; and 2) Kilifi County Hospital, a community-based government hospital serving a rural coastal region. The samples were collected during the COVID-19 pandemic between June 2020 and August 2022. At the time of the study, SARS-CoV-2 transmission in Nairobi was higher than in Kilifi [33]. Participants were adults, aged ≥18 years old, recruited within seven days of positive diagnosis of COVID-19 by RT-PCR testing. Initial sampling was at day 0 (i.e., day of diagnosis). Follow-up and subsequent samplings were done on days 7, 14 and 28 from a positive SARS-CoV-2 infection diagnosis. We collected 20 mL of venous blood in sodium heparin vacutainer. Additionally, we included residual longitudinal plasma samples from the AKUH biobank for cytokine and chemokine measurements. These samples were collected from COVID-19 patients who consented to this follow-up study on the day of diagnosis (i.e., day 0) and on day 28.

## COVID-19 severity classification

We included patients from five COVID-19 severity groups of asymptomatic, mild, moderate, severe, and critical as determined by clinicians at the time of diagnosis following the National Institutes of Health (NIH, USA) guidelines [34]. Asymptomatic patients were those who tested positive for SARS-CoV-2 via RT-PCR but did not display any COVID-19 symptoms. Mild cases were SARS-CoV-2 positive and exhibited symptoms such as fever, sore throat, cough, malaise, headache, muscle pain, vomiting, nausea, diarrhoea, anosmia, or ageusia, without any signs of shortness of breath, dyspnea, or abnormal chest imaging. Moderate cases tested positive for SARS-CoV-2 and showed evidence of lower respiratory tract infection based on clinical examination or imaging, with an oxygen saturation (SpO2) of ≥94% on room air. Severe cases were positive for SARS-CoV-2 with an SpO2 of <94% on room air, a ratio of arterial partial pressure of oxygen to fraction of inspired oxygen (PaO2/FiO2) <300 mm Hg, a respiratory rate >30 breaths/minute, and/or lung infiltrates >50%. Critically ill patients were positive for SARS-CoV-2 and experienced respiratory failure, septic shock, and/or multiple organ dysfunction syndrome. Due to the small numbers in the moderate and critical groups, mild cases were lumped together with moderate (mild/moderate), whilst critical were grouped with severe ones (severe). Thus, we studied immune responses among three COVID-19 patient groups: asymptomatic, mild/moderate and severe.

## Procedures

**Plasma separation and PBMC isolation.** Plasma was separated by centrifuging the tubes at 440 g for 10 minutes using an Eppendorf 5810R centrifuge, aliquoted in 2 mL microcentrifuge tubes, and immediately stored in −80°C freezers until the time for laboratory analysis. PBMCs were isolated from the remaining blood component using density gradient centrifugation media (Lymphoprep™ (1.077 g/ml, Stem Cell Technologies), aliquoted to 1.8 mL cryovials, and stored in −196°C liquid nitrogen tanks until usage. Plasma and PBMC samples from AKUH were transported in dry ice and liquid nitrogen respectively, to the KEMRI–Wellcome Trust Research Programme laboratories in Kilifi and stored appropriately for laboratory analyses. Prior to use, PBMCs were thawed and rested at 37°C, 5% $CO_2$ for 15–16 hours. PBMC counting was done using Vi-CELL XR Cell Viability Analyser or Countess™ Cell Counting Chamber Slides (Thermo Scientific) before assay setup.

**SARS-CoV-2 synthetic peptides pools for ELISpot measurements.** A total of 641 peptides (15–18-mers with a ten amino acid overlap) were pooled into ten peptide pools, spanning different regions of the virus. The pools covered: the spike protein region (S1: positions 1–93 and S2: positions 94–178), membrane protein (M: positions 1–31), nucleocapsid protein (NP: positions 1–55), non-structural proteins (NSP 3B: positions 207–306, NSP 3C: position 307–379, NSP 12B: positions 665–729 and NSP 15–16: positions 886–972) and Open Reading Frame (ORF3: positions 1–37 and ORF8: positions 1–15). These peptides (which were synthesised by Mimotopes Pty Ltd and a kind donation from Professor Susanna Dunachie's laboratory, Oxford University) were used to stimulate PBMCs for *ex vivo* IFN-γ ELISpot assay. The peptide sequences and pooling details are provided in S1 Table and S2 Table.

Because of limitations of PBMC numbers in individual samples, we prioritized performing the IFN-γ ELISpot assay against the top 10 of 22 peptide pools, based on the magnitude of the responses they induced in a pre-screening optimization experiment. The 10 pools that elicited the highest IFN-γ responses are S2, S1, NP, M, ORF3, NSP 12B, NSP 3C, NSP 15 + 16, NSP 3B and ORF8.

**Interferon gamma ELISpot assay.** To quantify *the ex vivo* interferon gamma (IFN-γ) cellular response to overlapping SARS-CoV-2 peptides from different proteins, we stimulated PBMC with synthetic peptides pools through an *in vitro* IFN-γ ELISpot assay, as previously described [35]. Briefly, 5 μg/ml of anti- human IFN-γ antibody clone 1-D1K (Mabtech, AB, Sweden) was used to coat Multiscreen-I 96 ELISpot plates overnight. Individual's PBMC were plated in duplicates at 200,000 cells per well for each specific protein, based on pre-prepared plate templates. Peptide pools were then added at a final concentration of 2 μg/mL per wells and incubated for 16–18 hours at $37^0$C, 5% $CO_2$, 95% humidity. Concanavalin A (ConA; Sigma) was used as the positive control at a final concentration of 5 μg/mL per well, while

dimethyl sulfoxide (DMSO; Sigma), which was a constituent of the diluent for the peptides and Con A was used at a similar concentration to peptides to serve as the negative control. IFN-γ secreting cells were then detected using an anti-human IFN-γ biotinylated antibody clone 7-B6-1 (Mabtech) at 1 μg/mL and an incubation for 2–4 hours. Thereafter, streptavidin alkaline phosphatase antibody (Mabtech) was added at 1 μg/mL and incubated for 1–2 hours, and the IFN-γ spots then developed using 1-Step™ NBT/BCI (nitro blue tetrazolium/5-bromo-4-chloro-3-phosphatase) substrate (Thermo Scientific) during a 7-minute incubation in the dark. The enzyme-substrate reaction was stopped by rinsing the plate 3 times under running tap water. Plates were then airdried for at least 2 days on an open lab bench, and spots enumerated on an AID ELISpot Reader version 4.0. Results are hereby reported as spot-forming units (SFU)/10^6 PBMC after subtracting the background (mean SFU from negative control wells). Data from failed individual PBMC tests, defined here as either, an excessive background where the negative control wells had > 80 SFU/10^6 PBMCs, or a positive control well with an average of <100 SFU/10^6 PBMCs (too few), were excluded. We also applied the ELISpot assay limit of detection of 10 SFU/10^6 PBMCs, with all wells having values <10 SFU/10^6 PBMCs replaced with 5 SFU/10^6 PBMCs. For the samples that did not have enough PBMC to be tested against all the peptide pools, we prioritised measurements against pools from S2, S1, NP and M proteins, as they induced the highest IFN-γ responses in our pilot screening experiment. Data are reported only for the individuals whose PBMC were tested against all the available peptide pools for each specific protein segment. We summed the responses from the different pools of the same protein segment, which resulted in different sample sizes for different proteins as follows: spike (171 samples for S1, S2), NP (162 samples), M (136 samples), NSPs (100 samples for NSP 3B, NSP 3C, NSP 12B and NSP 15–16) and ORF (90 samples for ORF 3 and ORF 8) S3 Table.

**Quantification of cytokines by Luminex assay.** Plasma concentrations of 22 cytokines and chemokines were measured using a Human ProcartaPlex™ Human Panel 1A (Thermo Fisher Scientific, Cat. No. EPX010-12010-901, Lot number 316776−000), which consisted of: a) Th1/Th2 specific cytokines: GM-CSF, IFN-γ, IL-1β, IL-2, IL-6, IL-8, IL-18, TNF-α, IL-9, IL-21; b) Pro-Inflammatory cytokines: IFN-α, IL-1α, IL-1ra, IL-7, TNF-β; and c) Chemokines: Eotaxin, GRO-α, IP-10, MCP-1, MIP-1α, MIP-1β, SDF-1α, according to the manufacturer's instructions. Briefly,1x capture magnetic beads were added to the plates, and unbound beads were then washed away with 1X wash buffer. Plasma samples were thawed and diluted at 1:2 with 1X universal assay buffer (UAB) before addition to the plates. Standards from the kit at 4-fold serial dilutions of 1:5, 1:20, 1:80, 1:320, 1:1280, 1:5120, 1:20480 as well as a blank (UAB), were also added. The plates were then incubated on a shaker at 600 rpm for 2 hours at room temperature. After incubation, contents were discarded, plates washed, and 1X biotinylated detection antibody added. The plates were then incubated on the shaker for 30 minutes. Upon washing, 1X Streptavidin-PE was added and incubated on the shaker for 30 minutes. The plates were then washed before adding 1X reading buffer and incubated on the shaker for 5 minutes. All incubations were done at room temperature, and the washing steps performed using an Invitrogen hand-held magnetic plate washer. Data were acquired on the Magpix systems multiplex Luminex machine and concentrations (pg/mL) of the samples calculated in Belysa® Immunoassay Curve Fitting software version 1.1.0 [36] using a 5- or 4-parameter logistic standard curve generated from standards of known concentration.

## Statistical analysis

For the ELISpot data, time point specific geometric means (GMs) of IFN-γ secreting cells for each of the different SARS-COV-2 peptides pools were calculated for each severity group and geographical location. For each peptide, variations in ELISpot responses were compared using a linear mixed effects model [37] on log-transformed PBMCs values with patient as a random effect, and time (i.e., day of sampling), severity group and time-by-severity group interaction term as fixed effects, followed by Tukey's multiple comparisons. Within each severity group, differences between geographic locations were compared using Kruskal Wallis paired with a Dunn's multiple comparisons. This analysis set was restricted to 59 patients who had PBMC samples.

Cytokine and chemokine data were first normalised to have a zero mean and a standard deviation of one, and cross-correlations among the cytokines determined using Pearson correlations and principal component analysis (PCA). Principal components (PCs) were extracted based on scree-plot, variance explained and the interpretability of the components. Next, the non-normalised cytokine and chemokine data were log-transformed and fitted into linear mixed effects regression (LMER) models with age, sex, day of sampling, location or disease severity as fixed effects and patient as a random effect. LMERs were used to exploit the hierarchical or clustered nature of our data [37]. Interactions were explored and separate models fitted where necessary. For location comparisons, severe and moderate COVID-19 cases were excluded as these categories were only present in the Nairobi data. Results from the regression models were presented using heat maps, in which the effect size (i.e., coefficient) determined the density of colour-shading for each square. TNF-β data were excluded from the analyses as only one patient had a measurable concentration. Thus, we analysed 21 cytokines from all 188 patients.

A total of 98 (20%) data values of the expected 498 were missing due to a variety of reasons including design, participants moving out of the study area after hospital discharge, insufficient PBMC numbers or plasma volumes, and unavailability of some participants during scheduled study visits. However, after exploration of the missing data (see supporting information, S1 Fig and S4 Table) coupled with the likelihood approaches, the analyses are valid under the missing at random mechanism [38].

Analyses were performed using R version 4.3.0 [39]. The factoextra package [40] was used for PCA. For visualisations, the pheatmap [41] and ggplot2 [42] packages and GraphPad Prism Software version 10.1.2 [43] were used.

### Ethical approval

The study obtained ethical approval from the Kenya Medical Research Institute's Scientific and Ethics Review Unit (KEMRI SERU; protocol no. 4081) and the Aga Khan University, Nairobi, Institutional Ethics Review Committee (protocol no. 2020 IERC-55 V5 and 2020 IERC-135 V2). Written informed consent was obtained from all willing patients before their enrolment into the study.

## Results

### Participant baseline characteristics

Using the NIH clinical guidelines for grading COVID-19 severity of 188 patients, 27 (14%) were asymptomatic, 75 (40%) were mild/moderate and 86 (46%) were severe cases. Collectively, the 188 patients contributed 400 blood samples collected over the first month of diagnosis: day 0 (187 samples), day 7 (50 samples), day 14 (53 samples), and day 28 (110 samples). Fewer samples were collected on days 7 and 14 than days 0 and 28 mainly due to study design: that is, additional clinical residual plasma samples were collected at AKUH on days 0 and day 28. Of the 188 patients, 129 (69%) were male, 36 (19%) were from rural Kilifi and 152 (81%) from urban Nairobi (Table 1). Their median age at recruitment was 48 years (IQR 37–58), with disease severity increasing with age. All 86 patients with severe COVID-19 were from Nairobi as we were unable to recruit severe patients in Kilifi. Underlying comorbidities such as diabetes, HIV/AIDS, and hypertension were prevalent among the mild/moderate and severe COVID-19 groups. Two participants with severe disease died on day 7 and 28 respectively (Table 1).

### Increased frequencies of *ex vivo* spike specific IFN-γ secreting cells at the time of COVID-19 diagnosis are associated with asymptomatic SARS-CoV-2 infection and rural residency

Kinetics and levels of IFN-γ secreting cell responses were assessed in a subset of 59 participants with PBMC samples, contributing 172 samples. At day 0, the frequency of IFN-γ secreting cells to overlapping SARS-CoV-2 spike peptides was significantly higher in asymptomatic patients compared to severe patients (GMs: 117 [95% CI 71–194] vs 32 [95%

**Table 1. Participant demographic and clinical characteristics[†].**

| Characteristic | Asymptomatic (n = 27) | Mild/Moderate (n = 75) | Severe (n = 86) | p-value |
|---|---|---|---|---|
| Median age at recruitment (IQR), years | 40 (33, 55) | 46 (35, 54) | 53 (43, 60) | **<0.001** |
| Male sex | 15 (56%) | 46 (61%) | 68 (79%) | **0.015** |
| Location | | | | **<0.001** |
| Kilifi | 21 (78%) | 15 (20%) | 0 | |
| Nairobi | 6 (22%) | 60 (80%) | 86 (100%) | |
| Diabetic (yes) | 1 (7%) | 28 (42%) | 47 (63%) | **<0.001** |
| Hypertensive (yes) | 0 | 15 (22%) | 33 (44%) | **<0.001** |
| HIV positive (yes) | 0 | 5 (8%) | 4 (5%) | 0.7 |
| Had COPD (yes) | 0 | 0 | 1 (1%) | >0.9 |
| Had Renal disease (yes) | 0 | 0 | 2 (3%) | 0.6 |
| Had heart disease (yes) | 0 | 0 | 6 (8%) | 0.081 |
| Were hospital admissions (yes) | 0 | 56 (84%) | 75 (100%) | **<0.001** |
| In intensive care unit (yes) | 0 | 4 (6.0%) | 15 (20%) | **0.018** |
| Clinical Outcome, died | 0 | 0 | 2 (3%) | 0.5 |

[†]Data are median (IQR) or number (%); COPD is Chronic obstructive pulmonary disease. P-values were calculated using Kruskal–Wallis for continuous variables and Chi-squared or Fisher's exact tests for categorical variables. Significance was set at p<0.05.

CI 8–76]; p = 0.0366) (Fig 1a, S5 Table). For the IFN-γ secreting cells specific to overlapping M peptides, severe patients exhibited significantly higher levels at day 7 than mild/moderate patients (GMs: 90 [95% CI 37–217] vs 19 [95% CI 10–39]; p = 0.0180; Fig 1b, S5 Table). We did not observe significant differences in the frequencies of IFN-γ secreting cell responses to NP, NSPs, and ORFs overlapping peptides at any of the time points (Fig 1c–1e, S5 Table).

For all five SARS-CoV-2 peptide pools we observed temporal variations in IFN-γ secreting cells within each COVID-19 severity group (Fig 1, S5 Table). Spike peptides: In the asymptomatic group, frequencies of IFN-γ secreting cells were significantly elevated on day 7 than on day 14 (GMs: 161 [95% CI 86–301] vs 92 [95% CI 44–189]; p = 0.0039) and day 28 (102 [95% CI 49–209]; p = 0.0042). For the mild/moderate group, levels of IFN-γ secreting cells significantly increased from 86 (95% CI 53–138) on day 7– 142 (95% CI 78–257) on day 14 (p = 0.0403). In the severe group, significantly higher levels of IFN-γ secreting cells were observed on day 7 (70 [95% CI 29–165]; p = 0.0381), day 14 (117 [95% CI 52–260]; p = 0.0058), and day 28 (108 [95% CI 33–348]; p = 0.0037) relative to day 0 (23 95% CI 7–76).

M Peptides: In the asymptomatic patients, IFN-γ secreting cells were significantly increased on day 14 (89 [95% CI 30–258]) compared to days 0 (GMs: 28 [95% CI 13–59]; p = 0.0364) and 7 (GMs: 48 [95% CI 18–129]; p = 0.0297). In the mild/moderate group, day 14 levels were significantly elevated compared to day 0 (p = 0.0039), day 7 (p < 0.0001), and day 28 (p = 0.002). Additionally, day 28 levels were higher than day 7 (p = 0.0049). No significant differences were observed across timepoints in the severe group.

NP Peptides: Among the asymptomatic group, a significant decline was observed between days 7 (71 [95% CI 33–153]) and 28 (47 [95% CI 24–88]; p = 0.0035). In the mild/moderate group, IFN-γ secreting cell levels were significantly increased on day 14 (GMs: 67 [95% CI 34–130]) compared to day 0 (GMs: 35 [95% CI 14–83]; p = 0.0037), day 7 (GMs: 44 [95% CI 26–73]; p = 0.0178), and day 28 (GMs: 47 [95% CI 24–91]; p = 0.0003). No significant differences were observed across timepoints in the severe group. For NSP peptides: a significant decline in IFN-γ secreting cells was observed within the mild/moderate group from 80 (95% CI 33–194) at day 14– 28 (95% CI 7–106) at day 28 (p = 0.0316). For ORF peptides: a significant increase in IFN-γ secreting cells was observed in the mild/moderate group between day 0 (14 [95% CI 4–43]) and day 7 (GMs: 38 [95% CI 16–85]; p = 0.0242).

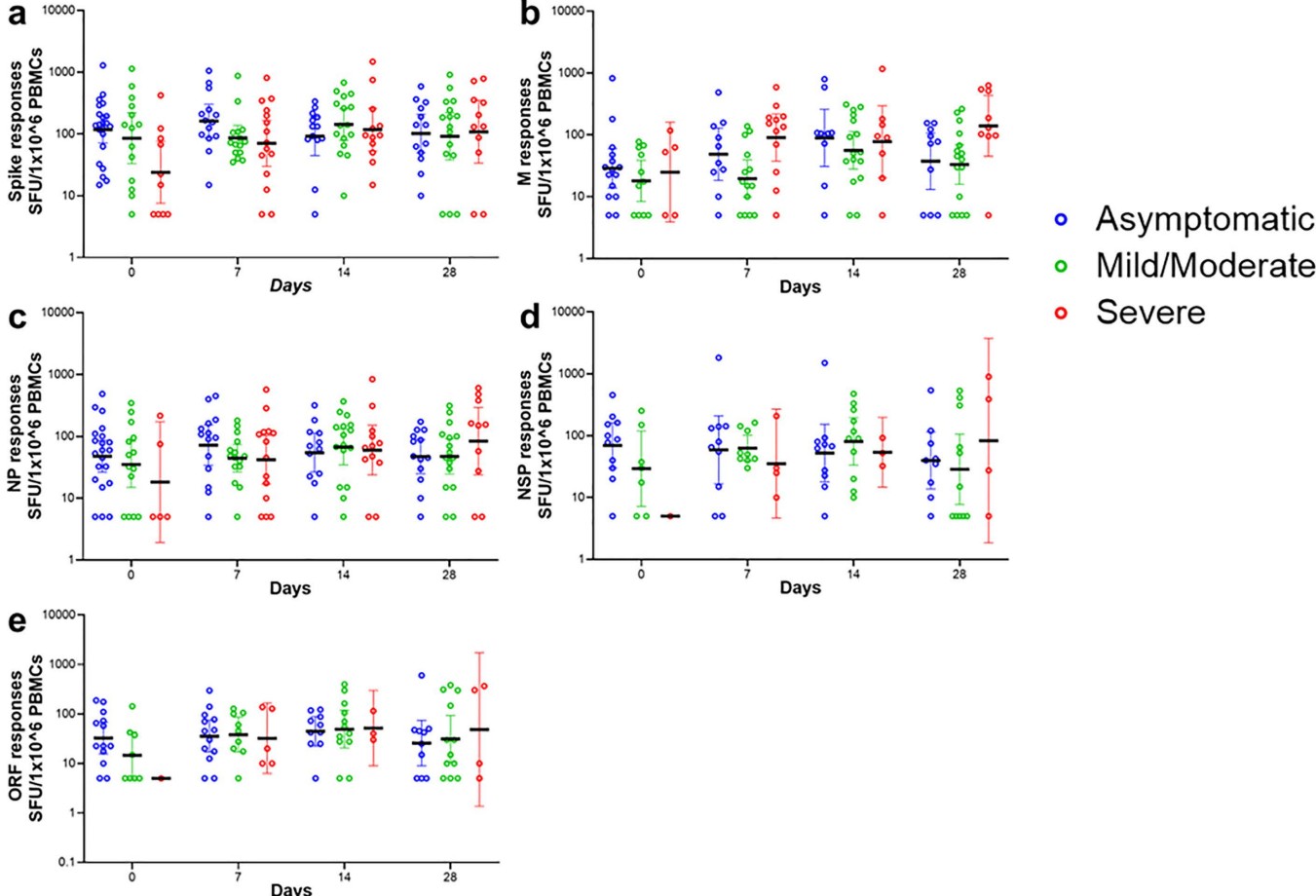

**Fig 1. Higher IFN-γ responses to spike protein were observed in asymptomatic infection at the day diagnosis.** Frequencies of ex-vivo IFN-γ secreting cells against SARS-CoV-2 peptide pools spanning (**a**) spike responses, (**b**) M responses, (**c**) NP responses, (**d**) NSP responses and (**e**) ORF responses. Bars represent geometric mean and 95% CI. Linear mixed effects model with Tukey's multiple comparisons, was used, * P < 0.05. Number of samples analysed for: spike responses = 171, M responses = 136, NP responses = 162, NSP responses = 100 and ORF responses = 90.

We also assessed whether there were differences in IFN-γ secreting cell responses by location (urban Nairobi versus rural Kilifi) among asymptomatic and mild patients. Relative to Nairobi, significantly higher levels of IFN-γ secreting cell responses to the SARS-CoV-2 spike peptides (218 [95% CI 125–381] vs 56 [95% CI 28–110]; p = 0.0057; Fig 2a) and NP peptides (94 [95% CI 51–169] vs 19 [95% CI 8–48; p = 0.0171; Fig 2c) were observed among the asymptomatic patients in Kilifi on day 0. There were no significant differences between Kilifi and Nairobi in the IFN-γ secreting cell responses to overlapping peptides for the M protein for the asymptomatic patients (Fig 2b), nor for the spike, or NP, and or M peptides among mild patients (Fig 2d–2f). Data for severe patients are shown for Nairobi participants only (Fig 2g–2i), as we were unable to recruit severe patients in Kilifi.

In summary, asymptomatic patients had higher anti-SARS-CoV-2 spike IFN-γ secreting cells than the symptomatic clinical phenotypes at the time of COVID-19 diagnosis, suggesting a possible protective role. Additionally, the asymptomatic individuals from rural areas had higher frequencies of anti-SARS-CoV-2 spike and NP peptides at the time of diagnosis than their counterparts in Nairobi city, suggesting modulation by either environmental exposures, or pre-existing cross-reactive protection.

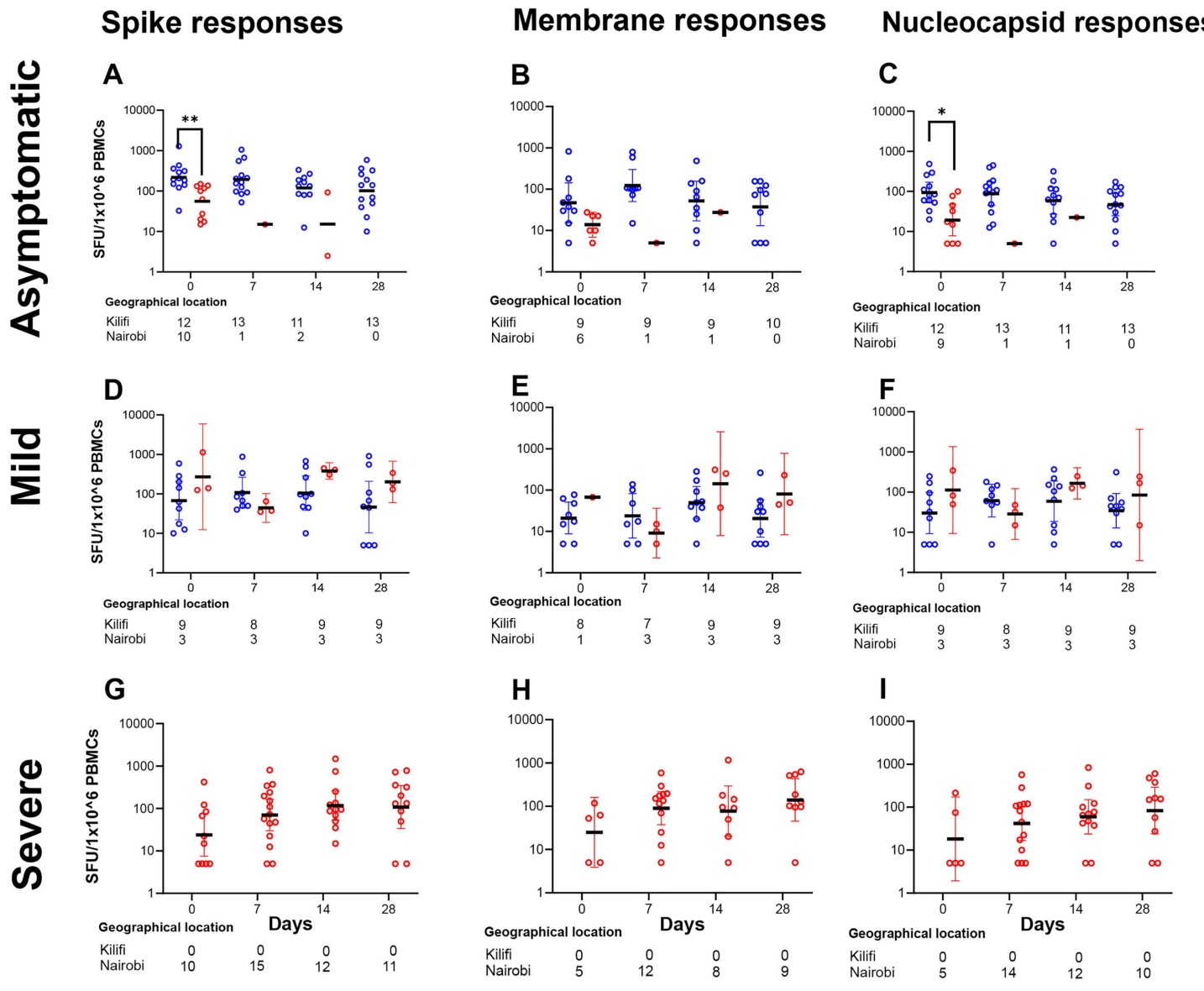

**Fig 2. Higher IFN-γ responses to spike protein were observed in asymptomatic infections from Kilifi than Nairobi at the day diagnosis.** Comparison of IFN-γ cellular responses between Kilifi and Nairobi COVID-19 patients with: asymptomatic disease for (**a**) Spike responses, (**b**) M responses, (**c**) NP responses; Mild disease for (**d**) Spike responses, (**e**) M responses, (**f**) NP responses; and Severe disease for (**g**) Spike responses, (**h**) M responses, (**i**) NP responses. Kilifi didn't have severe patients. Bars represent geometric mean and 95% CI. Kruskal–Wallis one-way ANOVA, with Dunn's multiple comparisons test, was performed. * P < 0.05, **P < 0.01.

## High plasma levels for SDF-1α are inversely associated with COVID-19 severity and rural residence

Asymptomatic patients consistently showed elevated levels of SDF-1α at all time-points, but lower levels of all the other cytokines and chemokines measured, and no detectable levels of IL-9 (Fig 3a). Similarly, for mild/moderate patients, high levels of SDF-1α (Fig 3b) were observed at all timepoints, whereas all the other cytokines and chemokines were secreted at low to intermediate levels. High cytokine and chemokine levels were seen among the severe patients with IL-Iβ, IL-6, IL-2, IFN-γ, GM-CSF, IFN-α, IL-7 and GRO-α decreasing over time, while MIP-1α, MIP-Iβ, MCP-1, and

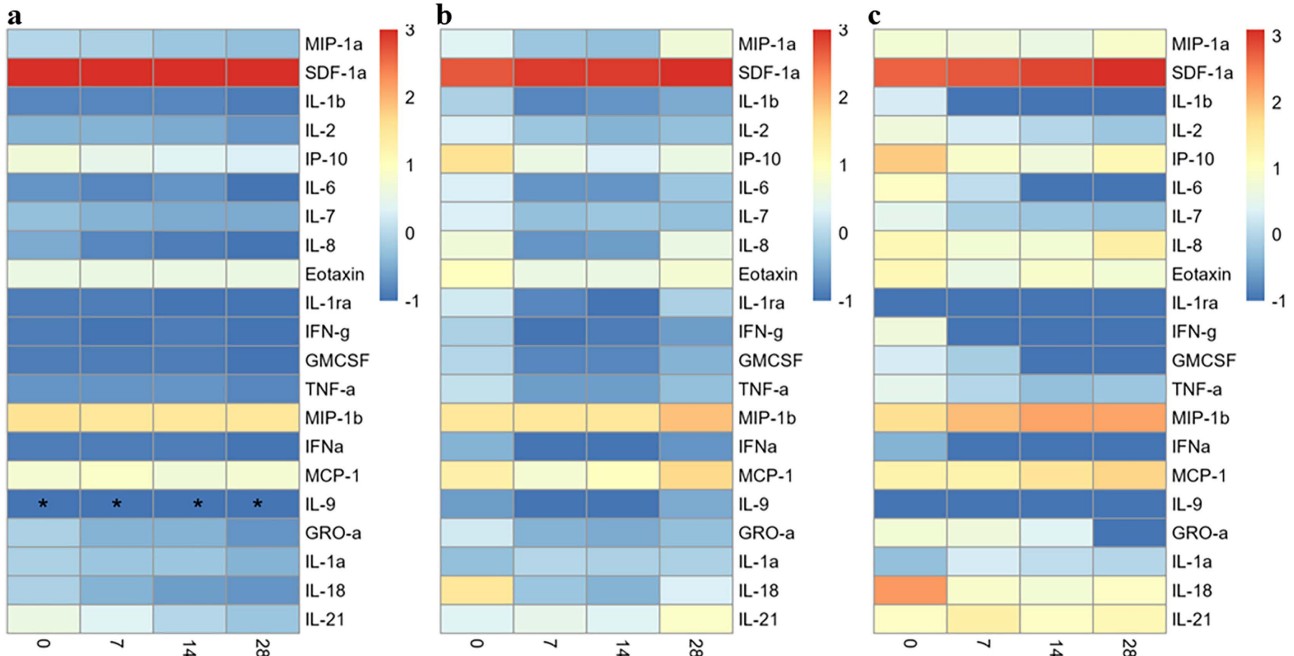

**Fig 3. High plasma levels of pro-inflammatory cytokines and SDF-1α are positively and inversely, respectively, associated with COVID-19 severity.** Mean of log10-transformed cytokine/chemokine concentrations plotted over time for **(a)** Asymptomatic participants, **(b)** Mild/Moderate participants, and **(c)** Severe participants. * - Levels for all participants were below detectable levels.

SDF-1α increased with time. On the other hand, levels for IL1-ra, and IL-9 were sustained at low levels at all the time points (Fig 3c).

We compared asymptomatic against mild/moderate and severe participants to evaluate differences among the clinical phenotypes. Mild/moderate participants had significantly increased levels of IL-18, IL-8, IL-1ra, IL-6, GM-CSF, IP-10, MCP-1, TNF-α, MIP-1α, IFN-γ, IL-2, IL-7, IL-1β, IL-9, Eotaxin and IFN-α than asymptomatic patients. The largest effect sizes were observed with IL-18 (1.107, p < 0.0001), IL-8 (1.105, p < 0.0001) and IL-1ra, (0.672, p = 0.013) (Fig 4). On the contrary, levels for SDF-1α were significantly reduced among the mild/moderate patients relative to the asymptomatic (effect size −0.182, p < 0.0001).

Severe participants had significantly increased levels for cytokines and chemokines (IL-8, IL-18, IL-1ra, IL-6, IP-10, MIP-1α, TNF-α, IL-9, IFN-γ, GM-CSF, IL-7, IL-1β, MCP-1, IL-2, GRO-α, Eotaxin and IFN-α) compared to asymptomatic cases, except for IL-1a, IL-21 and MIP-Iβ, which had similar levels, and SDF-1α (effect size −0.195, p < 0.0001), which was significantly reduced. The largest effect size was observed for IL-8 (1.754, p < 0.0001), IL-18 (1.666, p < 0.0001) and IL-1ra (1.197, p < 0.0001). Among the asymptomatic individuals, those from Nairobi had significantly elevated levels for IL-6, MIP-1α, IL-18, GRO-α, IL-2, IL-8, TNF-α and GM-CSF than Kilifi residents. Comparisons for IL-6 (1.139, p < 0.0001), MIP-1a (1.093, p = 0.004) and IL-18 (1.025, p = 0.002) had the largest effect sizes (Fig 4). For mild patients, all cytokines (IL-8, IL-18, IL-1ra, MIP-1α, IL-6, IP-10, IFN-γ, TNF-α, MCP-1, GM-CSF, IL-9, IL-1β, Eotaxin, IL-7, MIP-Iβ, and IFN-α) were elevated higher among Nairobi patients in comparison to Kilifi patients except for GRO-α, IL-1α, IL-2 and IL-21, which were similar (Fig 4). Notably, the largest effect size for these comparisons was observed for IL-8 (1.69, p < 0.0001), IL-18 (1.634, p < 0.0001) and IL-1ra (1.355, p < 0.001). On the other hand, SDF-1α (effect size −0.161, p = 0.017) was significantly lower in Nairobi than in Kilifi patients.

Collectively, the analyses above is consistent with COVID-19 immunology literature showing that high levels of pro-inflammatory cytokines are associated with severe disease. However, the association of high levels of inflammatory

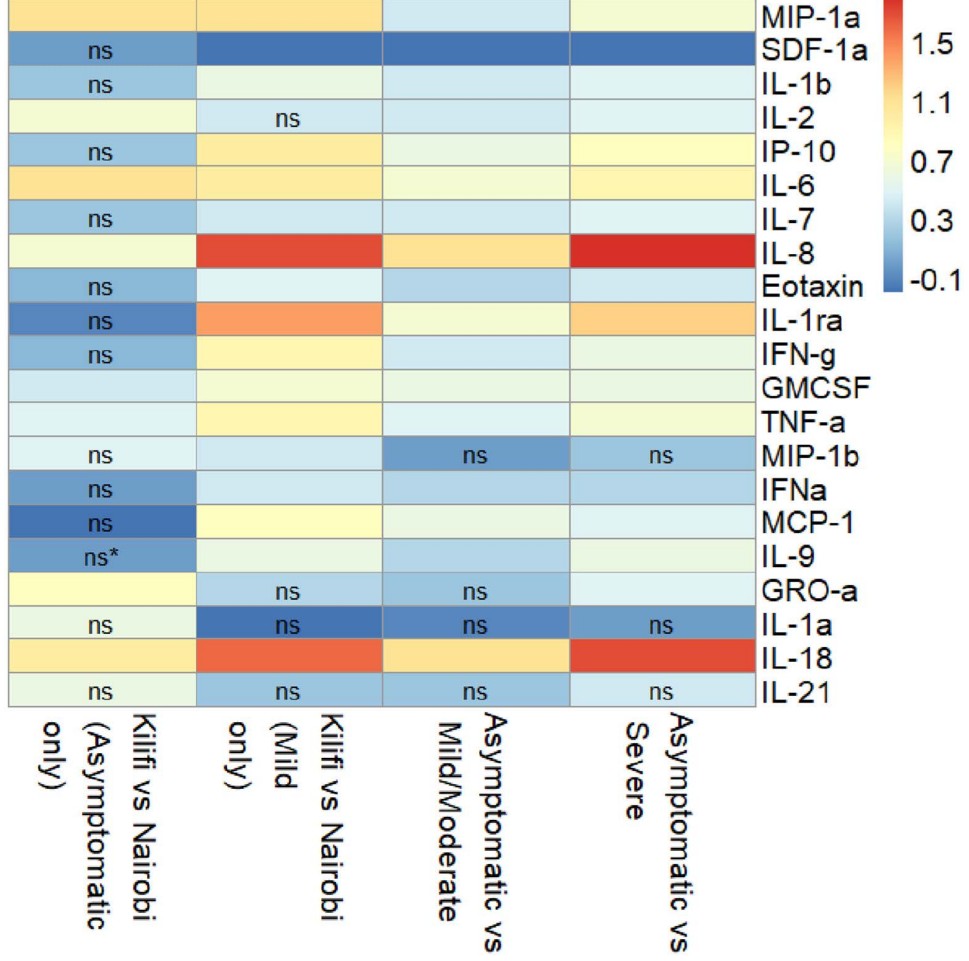

**Fig 4. Higher Cytokine and Chemokine concentrations in Asymptomatic and Mild Patients from Nairobi Compared to Kilifi, and in Mild/Moderate and Severe Patients Compared to Asymptomatic Individuals.** ns – not-significant, *p* value >0.05, empty means it was significant at p<0.05 −<0.0001. * - Levels for all participants were below detectable levels.

cytokines and chemokines with residency in Nairobi compared to Kilifi, and the vice versa for SDF-1α, would suggest modulation of the immune response by environmental exposures.

### Principal component analysis: clustering cytokine and chemokine responses by disease severity group

The concentrations of the majority of the 21 cytokines and chemokines were positively correlated with each other. However, the levels of SDF-1α were negatively correlated with those of IL-2, IP-10, IL-7, IFN-γ, GM-CSF, and IL-18 (S2 Fig). We retained the first three principal components from a principal component analysis, accounting for 62% of the total variability in the 21 cytokines and chemokines (S3a Fig). IL-9, MIP-1α, TNF-α, MCP-1, MIP-Iβ, IL1-ra, IL-6, IL-Iβ, GRO-α and IL-8 were the most strongly associated with the first PC. The second PC was most strongly associated with IP-10, IFN-γ, GM-CSF, IL-2, IL-18, IL-7, IFN-α, Eotaxin and SDF-1α. The third PC was associated with IL-1 α and IL-21 (S4 Fig). There was no clear separation among the mild/moderate and severe groups. However, data points for the asymptomatic participants clustered together demonstrating a much-reduced diversity of cytokine and chemokine levels than the other groups (Fig 5), and median PC scores did not change over the first month (S3b Fig).

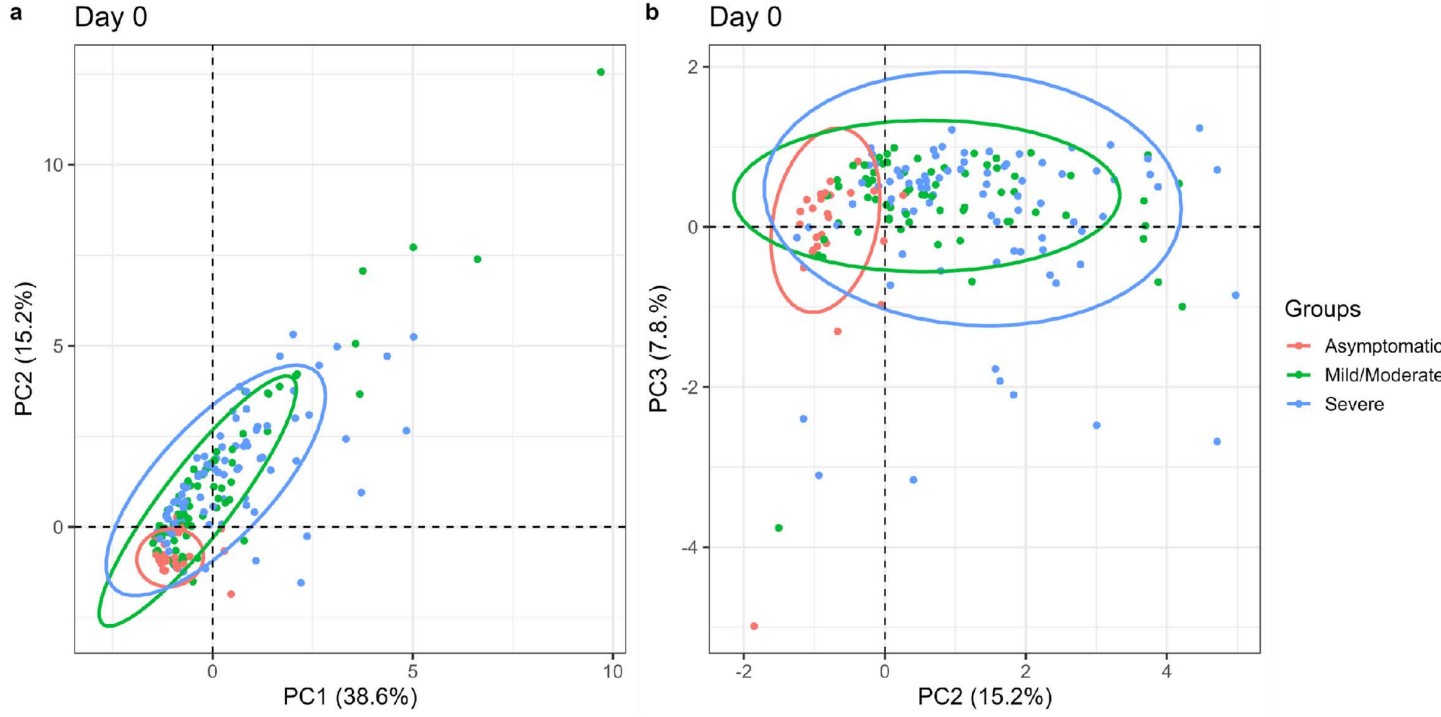

**Fig 5. Asymptomatic patients cluster together in scatter plots by Severity at day of diagnosis. (a)** PC1 vs PC2; **(b)** PC2 vs PC3.

Among the symptomatic groups (mild/moderate/severe), a steady decline of all the PCs over one month was observed, with a steep decline occurring between day 0 and day 7 (S3c Fig). There was no apparent difference of the cytokine and chemokine responses between asymptomatic participants from Kilifi and Nairobi (S5a Fig). For mild cases, Kilifi patients clustered together, although there was a slight overlap with Nairobi participants (S5b Fig). Using biplots, there was no apparent difference between asymptomatic and mild Kilifi patients (S5c Fig). For Nairobi, clinical phenotypes were asymptomatic, mild, moderate, and severe. We combined the mild and moderate, and observed no clear difference between the mild/moderate and severe groups, while the few asymptomatic participants seemed to cluster together (S5d Fig).

## Discussion and conclusion

In this study, we measured associations between the acute IFN-γ cellular, and 22 cytokine and chemokine, immune responses to SARS-CoV-2 (within a month of diagnosis) with COVID-19 severity and possible modulation by differential environmental exposures in urban and rural areas in Kenya. Our investigation revealed that a heightened cellular-IFN-γ response to SARS-CoV-2 spike peptides at the time of diagnosis is associated with asymptomatic infection, in marked opposition to both mild and severe presentations of COVID-19. The converse was true for SDF-1α, whose plasma levels were inversely associated with disease severity at all time points. We additionally observed divergent immune profiles when contrasting COVID-19 patients from urban and rural settings, with individuals from rural areas exhibiting diminished plasma concentrations of inflammatory markers but heightened plasma SDF-1α and early-cellular IFN-γ responses.

Our results corroborate earlier pandemic research performed in Brazil, Italy, the Netherlands, and Singapore, which reported a connection between increased levels of IFN-γ secreting cells at the onset of infection and the emergence of milder COVID-19 symptoms [44–47]. The finding suggests that, either: 1) the early IFN-γ cellular response contributes to protection against disease progression, or 2) that developing severe COVID-19 depresses the response. The latter

interpretation may be supported by previous reports from China and USA reporting correlations of an early induction of increased basal T-cell specific responses (CD4+ and CD8+ T) in COVID-19 patients with mild disease, which was suppressed among their severely sick counterparts [48,49].

Whilst the concentrations of the pro-inflammatory cytokines and chemokines IL-6, IL-Iβ, GRO-α, IFN-γ, GM-CSF, IFN-α, IL-7 and IL-2 in severe patients declined to basal levels at day 28 from day of diagnosis, we found that the concentrations for the chemokines MIP-1α, MIP-Iβ, MCP-1, and SDF-1α were increasing with time. This is to be expected as cytokines and chemokines play different roles, and at different times of the immune response to infection. MIP-1α, MCP-1 and MIP-Iβ are chemoattractant and play key roles in the recruitments of leukocytes such as monocytes, T-cells, and neutrophils to the sites of infection [50,51]. Similarly increasing kinetics for MIP-1α and MCP-1 in severe and fatal COVID-19 cases were reported in Norway and China, respectively [52,53]. Notably, we found that higher concentrations of SDF-1α were associated with asymptomatic individuals, hinting at a potential protective role from severe disease progression. SDF-1α (CXCL12), is a chemokine involved in the recruitment of T-cells [54], CD34 + hematopoietic stem/progenitor cells [55], lymphocytes and monocytes [56] to the site of infection, further enhancing inflammation. In contrast, studies from China and Bulgaria did not observe significant differences in SDF-1α levels when comparing asymptomatic, mild, moderate, severe or fatal cases [20,22,53]. Nevertheless, alternative research has indirectly associated SDF-1α with the severity of disease, as evidenced by genetic association analyses conducted in a single-center investigation [57]; however, this observation was not validated in multi-center genome-wide association studies [58]. SDF-1α may play a role in recruiting immune cells to infection sites [54], potentially contributing to viral clearance.

We also demonstrate that elevated levels of eighteen cytokines and chemokines were associated with severe COVID-19, in agreement with previous reports from China, Bulgaria, Rwanda, Hong Kong, Kenya, Ireland, Egypt and Iran, which associated these markers to disease severity, adverse outcomes, severe lung injury and ARDS [19–24,31,59–61]. However, the associations with IL-8, IL-18 and IL-1ra, had the strongest effect sizes in the current study. IL-8, has been implicated in the activation and recruitment of neutrophils to the site of infection, thereby promoting inflammation [62]. IL-18 amplifies the immune response by inducing the production of IFN-γ by T-cells and natural killer cells [63]. Thus, IL-8 and IL-18 could amplify the excessive inflammation that characterizes severe COVID-19. On the contrary, IL1-ra is known to suppress the production of proinflammatory cytokines such as IL-1 and TNF-α [64], and probably helps mitigate the effects of excessive inflammation, thereby reducing tissue damage and associated mortality. As evidenced in prior research, the individuals afflicted with severe COVID-19 in this study exhibited a greater prevalence of comorbid conditions, including diabetes, hypertension, cardiovascular ailments, and respiratory disorders, in comparison to those presenting with asymptomatic infection or mild symptoms. These comorbidities may influence immune responses by modulating angiotensin-converting enzyme 2 (ACE2) expression, altering cytokine and chemokine profiles [65], and hence are potential confounders in this correlational analysis.

In conjunction with the diminished incidence of severe COVID-19 cases and related fatalities in SSA when contrasted with North America and Europe, both our research and that of others have identified elevated occurrences of severe COVID-19 in urban locales as opposed to rural regions [33]. While this disparity may partly be explained by elevated rates of SARS-CoV-2 transmission in densely populated urban centers, or by more comprehensive case reporting [33,66,67] among other possibilities, we contemplated whether there existed credible biological causes. We compared inflammatory cytokine and chemokine levels and found that the asymptomatic patients from Nairobi had higher levels of 8 cytokines and chemokines than their asymptomatic counterparts from Kilifi, with IL-6, IL-18 and MIP-1α, having the strongest associations. Similarly, levels of 16 cytokines and chemokines were higher among Nairobi-mild than Kilifi-mild COVID-19 patients, with IL-8, IL-18 and IL-1ra being the most differentially secreted. Moreover, the basal frequencies of IFN-γ secreting cells in asymptomatic patients from Nairobi, relative to those of their Kilifi asymptomatic counterparts, were relatively reduced. Collectively, these findings suggest that the immune response to SARS-CoV-2 is less inflammatory among residents of rural areas [32]. The factors underlying this observation may include differences in previous environmental exposures

such as air pollution and microbes, as well as differences in lifestyles including nutritional and behavioral factors [68–70]. Consequently, the immune systems of individuals residing in urban and rural environments may have been predisposed to exhibit divergent responses to SARS-CoV-2, with inhabitants of urban regions demonstrating a higher propensity to generate elevated concentrations of inflammatory cytokines, which are correlated with the severity of COVID-19.

Our study was faced with a few limitations. Firstly, our data are incomplete for some of the patients due to missed follow-ups, or due to unavailability of adequate numbers of PBMCs to quantify IFN-γ cellular responses to the full spectrum of the peptide pools corresponding to all the different SARS-CoV-2 proteins. These gaps resulted in missing data for certain timepoints or peptide stimulations, which may have reduced the statistical power or even biased some comparisons. However, our analyses are assumed valid under the missing at random mechanism given the likelihood approach [38]. Secondly, we had difficulties recruiting asymptomatic patients in Nairobi and were unable to recruit severe cases in Kilifi, resulting in an imbalanced distribution across clinical phenotypes and thus the sample size is relatively small. These limitations may affect the generalizability of our findings, particularly in comparisons of disease severity between the two geographic locations. Future studies should aim to include a more balanced and diverse sample set to improve representativeness.

In conclusion, while the occurrence of severe disease was relatively infrequent in Kenya, the inflammatory cytokine profile observed in patients experiencing severe COVID-19 closely resembles that of individuals diagnosed with severe COVID-19 in North America and Europe. An elevated early secretion of IFN-γ by immune cells, along with heightened concentrations of the chemokine, SDF-1α, were correlated with asymptomatic infections, thereby implying potential protective functions of these immune responses in mitigating the advancement of the disease. Furthermore, the differential profiles of cytokines, chemokines, and IFN-γ secreting cells, observed between patients from Nairobi and those from Kilifi, provide a compelling biological explanation for the increased incidence of severe COVID-19 in SSA urban centers, in contrast to their rural counterparts. Collectively, these observations yield valuable understanding regarding the immune responses that may confer protection against COVID-19, as well as their modulation by varying environmental exposures in geographically distinct regions. Nonetheless, additional studies are required to extend and replicate these important findings as they could inform future control of COVID-19 and empower the control of similar pandemics.

## Supporting information

**S1 Table. Peptides Sequences.** Each entry represents a distinct peptide along with its associated properties.
(DOCX)

**S2 Table. Pooling strategy.** The pooling strategy for the 10 peptide pools used.
(DOCX)

**S3 Table. Sample size for each peptide.** Number of participants sampled for each peptide across different clinical phenotypes (asymptomatic, mild/moderate, severe) and time points (Day 0, 7, 14, 28).
(DOCX)

**S4 Table. Association between dropout and baseline characteristics.**
(DOCX)

**S5 Table. Associations of SARS-CoV-2 Peptides (Spike, M, NP, NSP, and ORF) with asymptomatic, mild/moderate, and severe clinical phenotypes over time. Tukey's multiple comparisons test was used to assess differences within and between clinical phenotypes across different time points.** Bold p values are significant, $p < 0.05$.
(DOCX)

**S1 Fig. Sampling plan for the study and samples available for analysis.**
(TIF)

**S2 Fig. Correlation matrix of 21 cytokines across all time points from 400 patients. Pearson's correlation coefficients are visualised, with red indicating positive correlations, blue negative correlations, and white representing no correlation.**
(TIF)

**S3 Fig. Principal component analysis of the cytokines. a) Scree-plot showing that the first 3 principal components explained 62% of variability in the cytokines data for all participants; (b) Line plots for the first 3 principal component illustrating trends over time for asymptomatic participants; and (c) Line plots for the first 3 principal component illustrating trends over time for symptomatic (mild, moderate and severe) participants.**
(TIF)

**S4 Fig. Cytokines loading on the first three principal components (PC1–PC3).** The color scale indicates the loading value, with red indicating a higher positive loading, blue a higher negative loading and light-yellow minimal loading.
(TIF)

**S5 Fig. Scatter plots showing cytokine data for participants grouped by location and clinical phenotype. (a)** Asymptomatic participants, **(b)** mild cases, **(c)** participants from Kilifi, and **(d)** participants from Nairobi. Each point represents an individual's cytokine measurement, allowing for a visual assessment of cytokine variability across different groups based on location and clinical presentation.
(TIF)

## Acknowledgments

We thank the field workers, laboratory staff and healthcare workers involved in the longitudinal blood sampling. We appreciate all the study participants.

## Author contributions

**Conceptualization:** Shahin Sayed, Reena Shah, Mansoor Saleh, Lynette Isabella Ochola-Oyier, Abdirahman I. Abdi, Philip Bejon, Eunice W. Nduati, Francis M. Ndungu.

**Data curation:** perpetual Wanjiku, John Kimotho, Jedidah Mwacharo, Viviane Olouch, Ann Karanu, Jasmit Shah, Zaitun Nneka.

**Formal analysis:** perpetual Wanjiku, Benedict Orindi, John Kimotho, Christopher Maronga, Philip Bejon, Eunice W. Nduati, Francis M. Ndungu.

**Funding acquisition:** Francis M. Ndungu.

**Investigation:** perpetual Wanjiku, John Kimotho, Jedidah Mwacharo.

**Methodology:** perpetual Wanjiku, John Kimotho, Jedidah Mwacharo, Abdirahman I. Abdi, Susanna Dunachie, Eunice W. Nduati, Francis M. Ndungu.

**Project administration:** Shahin Sayed, Reena Shah, Francis M. Ndungu.

**Resources:** Susanna Dunachie, Francis M. Ndungu.

**Supervision:** Shahin Sayed, Reena Shah, Philip Bejon, Eunice W. Nduati, Francis M. Ndungu.

**Validation:** Francis M. Ndungu.

**Visualization:** perpetual Wanjiku, Benedict Orindi, John Kimotho, Christopher Maronga.

**Writing – original draft:** perpetual Wanjiku, Benedict Orindi.

**Writing – review & editing:** perpetual Wanjiku, Benedict Orindi, John Kimotho, Shahin Sayed, Reena Shah, Mansoor Saleh, Jedidah Mwacharo, Christopher Maronga, Viviane Olouch, Ann Karanu, Jasmit Shah, Zaitun Nneka, Lynette Isabella Ochola-Oyier, Abdirahman I. Abdi, Susanna Dunachie, Philip Bejon, Eunice W. Nduati, Francis M. Ndungu.

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
