## [Decision Letter · Decision Letter 0]

8 Apr 2025

PONE-D-24-57722Induction of an early IFN-γ cellular response and high plasma levels of SDF-1α are inversely associated with COVID-19 severity and residence in rural areas in Kenyan patientsPLOS ONE

Dear Dr. Wanjiku,

Thank you for submitting your manuscript to PLOS ONE. After careful consideration, we feel that it has merit but does not fully meet PLOS ONE’s publication criteria as it currently stands. Therefore, we invite you to submit a revised version of the manuscript that addresses the points raised during the review process.

We look forward to receiving your revised manuscript.

Kind regards,

Ewurama Dedea Ampadu Owusu, PhD

Academic Editor

PLOS ONE

Journal Requirements:

Additional Editor Comments:

Authors should pay particular attention to the comments by Reviewer 2 in their revisions.

Reviewers' comments:

Reviewer's Responses to Questions

**Comments to the Author**

1. Is the manuscript technically sound, and do the data support the conclusions?

Reviewer #1: Yes

Reviewer #2: Yes

2. Has the statistical analysis been performed appropriately and rigorously? 

Reviewer #1: Yes

Reviewer #2: Yes

3. Have the authors made all data underlying the findings in their manuscript fully available?

Reviewer #1: Yes

Reviewer #2: Yes

4. Is the manuscript presented in an intelligible fashion and written in standard English?

Reviewer #1: Yes

Reviewer #2: Yes

5. Review Comments to the Author

Reviewer #1: I thoroughly enjoyed reading the manuscript. It is well-written, with findings clearly articulated and thoughtfully discussed. Only minor corrections are needed, along with a few adjustments to meet the publication criteria.

1. The abstract exceeds the 300-word limit required by the publication criteria. Please revise it to ensure compliance.

2. A minor grammatical error in the introduction (line 82).

3. According to the publication criteria, the introduction must include a brief statement of the overall aim of the study and a comment on whether the aim was achieved. While the aim is stated, there is no mention of its achievement. Please revise.

4. The study design is robust, with well-defined patient groups and clear cytokine/chemokine measurement methods. However, can you clarify how the missing data was handled in statistical analysis?

5. The experiment was conducted rigorously, with appropriate controls and sample sizes. The conclusion appropriately summarizes the results, emphasizing the association between IFN-gamma response, cytokine levels and COVID-19 severity. However, I suggest you include a discussion on potential confounding variables such as comorbidities. This could strengthen the argument.

6. The discussion is well-structured, but I suggest integrating more comparisons with similar studies outside Sub-Saharan Africa.

7. Figures and tables effectively present the data but should be closely examined for consistency in labeling (example: units, statistical significance markers).

8. References are comprehensive, but a few citations should be updated to include recent studies.

Reviewer #2: Summary:

The study investigates the immunological factors that might explain the lower severity of COVID-19 in Sub-Saharan Africa (SSA) compared to Europe and North America. It focuses on the levels of ex vivo SARS-CoV-2 peptide-specific IFN-γ producing cells and plasma cytokines and chemokines over the first month of COVID-19 diagnosis among Kenyan patients from urban (Nairobi) and rural (Kilifi) areas.

Methods:

• Participants: 188 COVID-19 patients from Nairobi (urban, n = 152) and Kilifi (rural, n = 36) were recruited and monitored longitudinally.

• Assays: IFN-γ secreting cells were enumerated using an ex vivo enzyme-linked immunospot (ELISpot) assay, and levels of 22 plasma cytokines and chemokines were measured using a multiplexed binding assay.

Results:

• IFN-γ Response: Higher frequencies of IFN-γ-secreting cells against SARS-CoV-2 spike peptides were observed on the day of diagnosis among asymptomatic patients compared to those with severe COVID-19.

• Cytokines and Chemokines: Higher concentrations of 17 out of 22 cytokines and chemokines were positively associated with severe disease, particularly IL-8, IL-18, and IL-1ra, while lower concentrations of SDF-1α were associated with severe disease.

• Geographical Differences: Concentrations of 8 and 16 cytokines and chemokines, including IL-18, were higher among Nairobi asymptomatic and mild patients compared to their Kilifi counterparts. Conversely, SDF-1α concentrations were higher in rural Kilifi compared to Nairobi.

Conclusion:

Pro-inflammatory cytokines and chemokines were associated with severe COVID-19, while an early IFN-γ cellular response to overlapping SARS-CoV-2 spike peptides was associated with reduced risk of disease. Living in urban Nairobi was associated with increased levels of pro-inflammatory cytokines/chemokines compared to rural Kilifi.

Loopholes in the Results and Discussion:

1. Sample Size and Recruitment Issues: The study faced difficulties recruiting asymptomatic patients in Nairobi and severe cases in Kilifi, leading to a relatively small sample size. This limitation could affect the generalizability of the findings.

2. Missing Data: Some data are incomplete for certain patients due to missed follow-ups or unavailability of adequate numbers of PBMCs to quantify IFN-γ cellular responses to the full spectrum of peptide pools.

3. Geographical Differences: The study suggests that the immune response to SARS-CoV-2 is less inflammatory among residents of rural areas. However, the differences in cytokine and chemokine levels between urban and rural patients might also be influenced by other factors such as environmental exposures and lifestyle differences.

4. The study does not provide a background on cellular response data against COVID-19 data available in the Kenyan population.

Other comments are in the attached draft.

6. PLOS authors have the option to publish the peer review history of their article (what does this mean? ). If published, this will include your full peer review and any attached files.

**Do you want your identity to be public for this peer review?** For information about this choice, including consent withdrawal, please see our Privacy Policy .

Reviewer #1: **Yes: ** Elizabeth Obeng-Aboagye

Reviewer #2: No

---

## [Author Response · Author response to Decision Letter 1]

7 May 2025

Dear PLOS ONE editor and reviewers,

Please note I have uploaded a rebuttal letter addressing all points raised by the reviewers.

Regards,

Perpetual.

---

## [Decision Letter · Decision Letter 1]

2 Jul 2025

PONE-D-24-57722R1Induction of an early IFN-γ cellular response and high plasma levels of SDF-1α are inversely associated with COVID-19 severity and residence in rural areas in Kenyan patientsPLOS ONE

Dear Dr. Wanjiku,

Thank you for submitting your manuscript to PLOS ONE. After careful consideration, we feel that it has merit but does not fully meet PLOS ONE’s publication criteria as it currently stands. Therefore, we invite you to submit a revised version of the manuscript that addresses the points raised during the review process.

**ACADEMIC EDITOR: **

The revised version submitted by authors has strengthened the manuscript. However, the incorporation of these additional comments by the reviewer can make the manuscript ready for publication in Plos One. Authors should pay attention to the follow up questions/suggestions by the reviewer 2 in the attachment.

We look forward to receiving your revised manuscript.

Kind regards,

Ewurama Dedea Ampadu Owusu, PhD

Academic Editor

PLOS ONE

Journal Requirements:

Reviewers' comments:

Reviewer's Responses to Questions

**Comments to the Author**

1. If the authors have adequately addressed your comments raised in a previous round of review and you feel that this manuscript is now acceptable for publication, you may indicate that here to bypass the “Comments to the Author” section, enter your conflict of interest statement in the “Confidential to Editor” section, and submit your "Accept" recommendation.

Reviewer #2: (No Response)

2. Is the manuscript technically sound, and do the data support the conclusions?

Reviewer #2: Yes

3. Has the statistical analysis been performed appropriately and rigorously? 

Reviewer #2: Yes

4. Have the authors made all data underlying the findings in their manuscript fully available?

Reviewer #2: Yes

5. Is the manuscript presented in an intelligible fashion and written in standard English?

Reviewer #2: Yes

6. Review Comments to the Author

Reviewer #2: I thank the author’s for taking time to revise the manuscript. However, I do have some comments for the authors in the attached file.

7. PLOS authors have the option to publish the peer review history of their article (what does this mean? ). If published, this will include your full peer review and any attached files.

**Do you want your identity to be public for this peer review?** For information about this choice, including consent withdrawal, please see our Privacy Policy .

Reviewer #2: No

---

## [Author Response · Author response to Decision Letter 2]

12 Aug 2025

We have carefully addressed all the concerns and comments raised by reviewer 2, this has greatly improve our manuscript. Please let me know if I can be of any further assistance.

Regards,

Perpetual.

---

## [Editor Report · Decision Letter 2]

19 Aug 2025

Induction of an early IFN-γ cellular response and high plasma levels of SDF-1α are inversely associated with COVID-19 severity and residence in rural areas in Kenyan patients

PONE-D-24-57722R2

Dear Dr. Wanjiku,

We’re pleased to inform you that your manuscript has been judged scientifically suitable for publication and will be formally accepted for publication once it meets all outstanding technical requirements.

Kind regards,

Ewurama Dedea Ampadu Owusu, PhD

Academic Editor

PLOS ONE

Additional Editor Comments (optional):

Dear Authors,

Responses to reviewer comments are satisfactory and relevant revisions have been effected.

But in line 73, the sentence begins with the same first two words as the sentence immediately following it in line 77 - "However, the...". Authors should endeavor to correct it before publication as it is repetitive and affects the comprehension of the conjunction between the two sentences.
---

## [Editor Report · Acceptance letter]

PONE-D-24-57722R2

PLOS ONE

Dear Dr. Wanjiku,

I'm pleased to inform you that your manuscript has been deemed suitable for publication in PLOS ONE. Congratulations! Your manuscript is now being handed over to our production team.

Kind regards,

on behalf of

Dr. Ewurama Dedea Ampadu Owusu

Academic Editor

PLOS ONE